# Survival status and its predictors among multi-drug resistance tuberculosis treated patients in Ethiopia: Multicenter observational study

**Asnake Balche Bade[1], Teshale Ayele Mega[2]***

**1** School of Pharmacy, Institute of Health Science, Arba Minch University, Southern Nations, Nationalities and Peoples Region, Arba Minch, Ethiopia, **2** School of Pharmacy, College of Health Sciences, Addis Ababa University, Addis Ababa, Ethiopia

* tesh.ayu2016@gmail.com

## Abstract

### Background

Although substantial progress has been made in combating the crisis of multi-drug resistance tuberculosis (MDR-TB), it remained the major public health threat globally.

### Objective

To assess patients' survival and its predictors among patients receiving multi-drug resistance tuberculosis treatment at MDR-TB treatment centers of southern and southwestern Ethiopia.

### Methods

A multicenter retrospective observational study was conducted from April 14 to May 14, 2019, among patients receiving MDR-TB treatment at three MDR-TB treatment centers, Butajira, Arbaminch and Shenengibe Hospitals, located in south and southwestern Ethiopia. A total of 200 records were reviewed using a check list adopted from the national MDR-TB treatment charts and other relevant documents. Data were entered into Epi-Data version 4.2.0 for cleaning and exported to STATA-13 for analysis. Descriptive analysis was carried out and results were presented by text, tables, and charts. Kaplan-Meier (log-rank test) and Cox regression were used to compare baseline survival experience and to determine predictors of patients' survival (death), respectively. The adjusted hazard ratio (AHR) was used to measure the strength of association and a p-value of <0.05 was considered to declare statistical significance.

### Results

Of 200 patients, 108 (54%) of them were males. The mean (+ standard deviation) age of the study population was 32.9±9.5years. During follow-up, 22 (11%) deaths were reported. The overall incidence density of death was 11.99, 95% CI [7.89–18.21] per 100,000person-

**Funding:** The authors received no specific funding for this work.

**Competing interests:** The authors have declared that no competing interests exist.

**Abbreviations:** Cm, Capreomycin; E, Ethambutol; Eto, Ethionamide; MDR-TB, Multidrug resistance tuberculosis; PAS, Para-aminosalicylic acid; Pto, prothionamide; PZA, pyrazinamide; R, Rifampicin; RDT, Rapid diagnostic test; S, Streptomycin; XDR-TB, Extensively drug resistance tuberculosis.

years. The median (interquartile range (IQR)) survival time was 375(249–457) days. Comorbidity (AHR = 23.68, 95% CI [4.85–115.46]), alcohol consumption (AHR = 4.53, 95% CI [1.21–16.97]), and history of poor adherence (AHR = 12.27, 95% CI [2.83–53.21]) were independently associated with patients' survival (death).

## Conclusion

In this study, the incidence density of mortality was very high. Alcohol consumption, poor adherence, and the presence of comorbidity were independently associated with death. Hence, alcohol users, patients with comorbidity and poor adherence should be given due attention during therapy.

## Introduction

Multi-drug resistant tuberculosis (MDR-TB) is the major concern at global, regional and country levels. According to the 2019 global TB report, there were 3.4% new and 18% previously treated cases of MDR-TB in 2018. In Bangladesh, among reported MDR-TB cases, 1.5% of them were new and 4.9% of them were previously treated TB cases. The incident rate of MDR-TB cases in this region was 3.7%. In the Democratic Republic of Congo (DRC), 1.7% of new cand 9.5% of previously treated TB cases of MDR-TB were reported. The overall incidence of MDR-TB in DRC was found to be 7.2%. Ethiopia ranked 8[th] among the 30 high MDR-TB burden countries with 2,700 MDR-TB cases each year. The estimated prevalence of MDR-TB in the country is 0.71% among newly diagnosed patients and 16% in patients under re-treatment [1].

MDR-TB was also responsible for a sizeable number of TB-related deaths globally. A study from the United Kingdom reported the death rate of 6.4% [2]. According to *Peter et al*, 3.9% of the deaths were accounted for MDR-TB [3]. Studies from India and South Africa found the mortality rate of 17% [4] and 20%, respectively, due to MDR-TB [5]. In Tanzania, 6.5% of mortality was reported among MDR-TB patients [6]. Two studies from Ethiopia revealed a mortality rate of 24.4% [16] and 18.3% [7] among patients receiving MDR-TB treatment.

Evidences showed the mortality rate due to MDR-TB was largely amplified by the presence of comorbidities [8]. Of note, the human immune virus (HIV) co-infection was the major risk factor [6, 9]. In one study, 31.3% of patients co-infection with HIV have died at 12 months of follow-up. Similarly, moderate to severe anemia and being smear positive were also associated with death [10]. In Ethiopia, the mortality rate of MDR-TB patients was higher in the earlier stages of treatment. Complications, drug-resistance, and smoking had contributed to an increased risk of mortality [11]. Though fewer studies had explored the incidence of mortality (patients' survival) in some parts of Ethiopia, they were single centered and hence, difficult to conclude the true incidence of national mortality. Therefore, this multi-center study was aimed to assess the incidence (incidence density) of mortality and its predictors among patients receiving MDR-TB therapy at MDR-TB treatment centers located in the south and southwestern regions of the country.

## Methods

### Study design and setting

A multicenter retrospective observational study was conducted from April 14 to May 14, 2019, among patients receiving MDR-TB treatment at Butajira, Arbaminch and Shenengibe General

Hospitals, all located in south and southwestern part of Ethiopia. The study settings are about 113km, 505km, and 329km, respectively, away from Addis Ababa, the political center of Ethiopia.

Butajira General Hospital is located at the Gurage zone (southern Ethiopia) and currently serving around 5 million population. It started the MDR-TB treatment service in 2015. During the this study, the treatment center had registered 65 MDR-TB patients. Of which 49 patients had finished treatment and 16 patients were on treatment. Arbaminch General Hospital is located in the Gamo zone (southern Ethiopia) and currently serving around 6 million population. The Hospital started the MDR-TB treatment service in January 2014. Of 50 patients registered at this treatment center, 45 patients had finished treatment and 5 patients were on the treatment. Shenengibe General Hospital is located in the Jimma zone (southwestern Ethiopia) and it is serving nearly 5 million people. It started the MDR-TB treatment service by January 2013. Since then, 98 MDR-TB patients were enrolled in to the TB treatment program and those who completed treatment and currently on treatment were 63 and 35 patients, respectively.

## Study population and patient enrollment

All adult MDR-TB patient charts who fulfilled the eligibility criteria at the aforementioned health care facilities were consecutively enrolled in to the study. Moreover, adult patients with Xpert MTB/RIF® assay [12] confirmed diagnosis of MDR-TB, enrolled in to the MDR-TB treatment program since January 2013, and whose charts contained complete baseline and follow-up data were included. Charts containing any degree of missed baseline and follow-up data (mortality and treatment response variable) as well as charts of patients transferred to other facilities were excluded. Finally, 213 charts were assessed for eligibility and 200cahrts were included in the final analysis (Fig 1).

## Data collection procedures and study variables

The data was collected by using a checklist prepared from different literatures, world health organization (WHO) MDR-TB treatment guidelines [13–15] and national MDR-TB treatment follow up chart. The checklist contains the following variables; patient-related variables such as; age, sex, residence, pregnancy status, marital status, smoking status, educational level, height, weight, and body mass index (BMI). The disease-related variables include category of MDR-TB, drug resistance status, and comorbidities. Furthermore, the checklist also contained drug-related variables such as; type of medication and drug regimen. The time of treatment initiation and the point at which mortality occurred was also recorded to calculate the median survival time to event (death). All the above data were extracted from patient charts. The study was approved by the Ethical Review Board of Jimma University and given an IRB number of IHRPG1/565/2019.

## The outcome definition and measurment

Mortality (death) due to MDR-TB complications or MDR-TB treatment related and the median survival time to mortality were considered as the outcome variables. The incidence (incidence density) of mortality was reported as person-years and the median survival time to mortality was reported in days. The overall median survival time and the median survival time corresponding to each MDR-TB treatment regimens was reported (Fig 2). Mortality data was obtained from the patients' discharge summary notes. Patients' mortality was confirmed by the signature of the caring physician, authorized by the service providing institution.

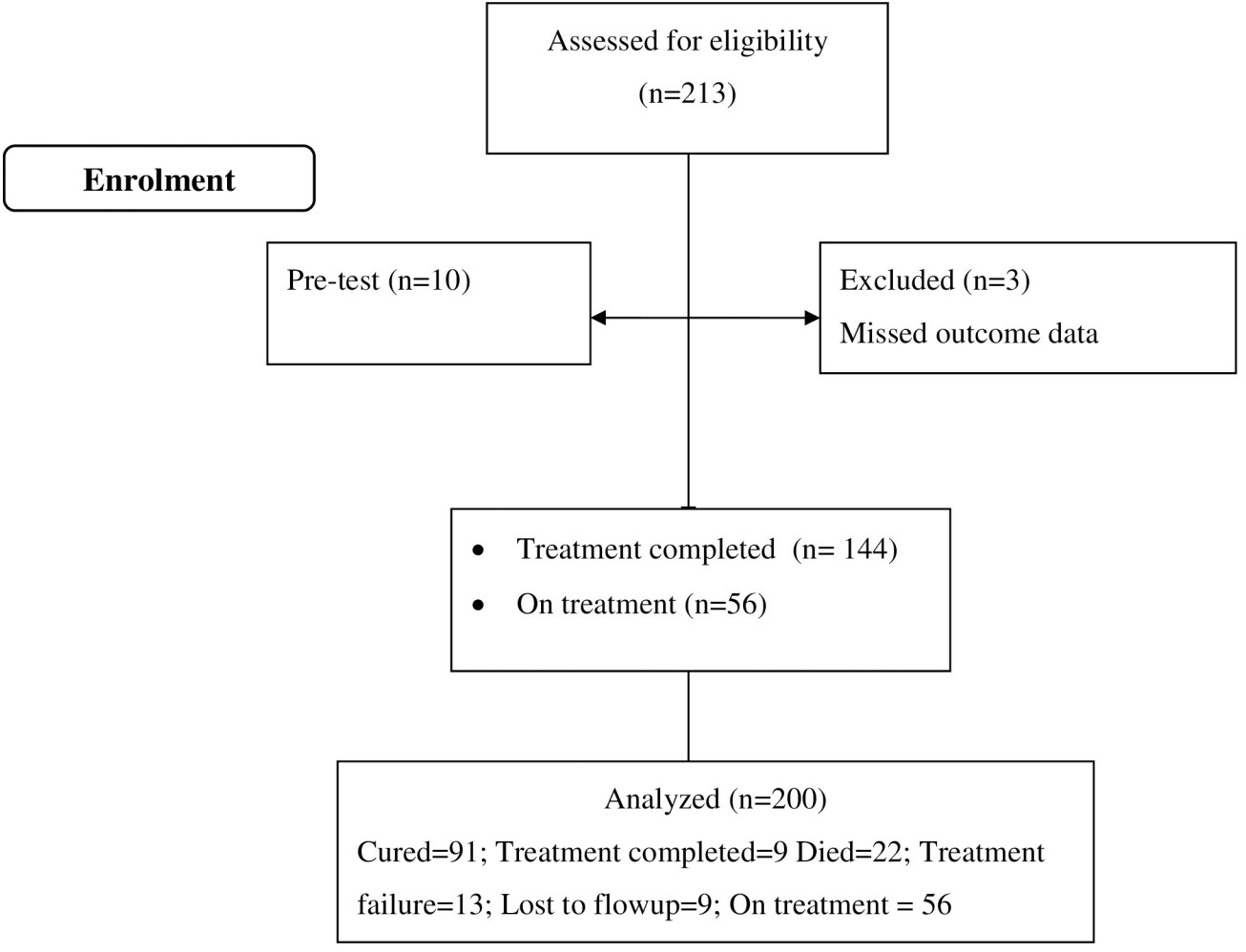

**Fig 1. Sample recruitment chart of patients who received MDR-TB treatment at MDR-TB treatment centers located at South and Southwest Ethiopia, April 14 to May 14, 2019.**

## Data quality assurance

The data collection tool was carefully designed to capture all necessary variables to achieve the study objectives. Each patient charts were reviewed for inclusion before the data collection. Three clinical pharmacists and three physicians were trained for two days on the contents of data collection tool and general procedures. The clinical pharmacists collected drug-related information and the patient-related and clinical variables were collected by the physicians. The clinical pharmacists were also responsible to identify and cross-check adverse drug reactions associated with each anti-TB drugs. At each facility, a senior infectious disease specialist supervised the overall activities including the data collection process. The supervisors were also responsible to ensure the diagnostic and clinical findings were truly related with the main outcome. Moreover, a pre-test was conducted on 5% of patients' records to test the effectiveness of the data collection tool and the necessary adjustment was made based on the pre-test findings.

## Data processing and analysis

The data were checked for completeness, cleaned with Epi-Data version 4.2 and exported to STAT-13 for analysis. Categorical variables were summarized by counts, graphs and

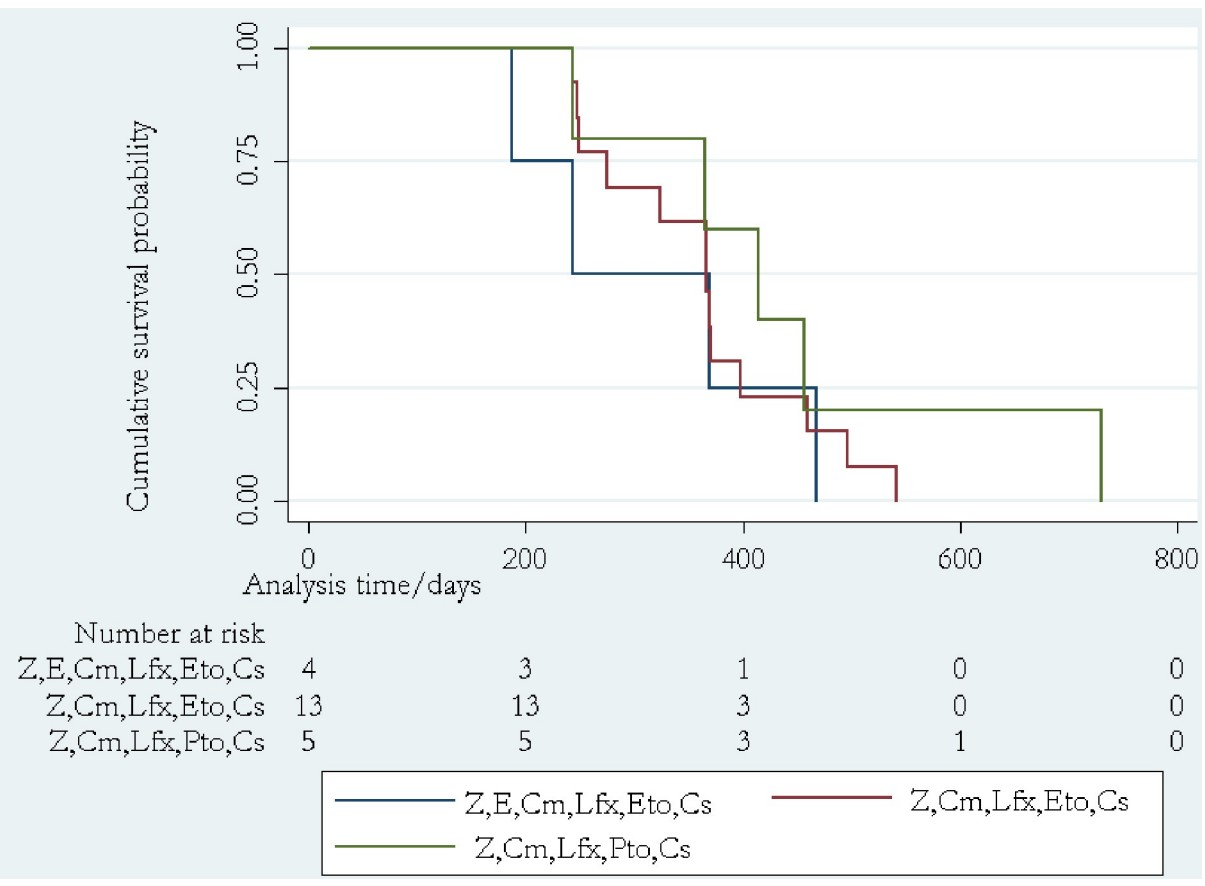

**Fig 2. The cumulative survival probability of MDR-TB treated patients with respect to the initial drug regimens at MDR-TB treatment centers of South and Southwest Ethiopia, April 14 to May 14, 2019.**

percentages. The baseline characteristics of the patients were compared using chi-square (χ2) test. Normally distributed continuous variables were reported using mean (+standard deviation (SD)), whereas median (interquartile range (IQR)) was utilized to describe non-normally distributed continuous variables. The study outcomes (mortality and time to death) were described as person-years and median (IQR) survival times, respectively. The base-line survival experience of the patients was estimated using the Kaplan–Meier (log-rank test) curve. The Cox regression model assumption of proportional hazards was checked by testing the interaction of covariates with time. Bivariate Cox regression was performed to select variables for multivariable Cox regression. Variables with p-value < 0.25 in bivariate Cox regression were considered for multivariable Cox regression. Multivariate Cox regression was performed to identify independent predictors of patient survival (mortality). The hazard ratio (HR) was used as a measure of the strength of association and p-value < 0.05 was considered to declare statistical significance.

## Operational definition

**Adherence.** It was calculated by dividing the missed doses by the totally prescribed dose and multiplied by 100. Therefore, it is described as Good (G) for > 95%, Fair (F) for 85–94% and Poor (P) for < 85% level of adherence based on patient's self report.

**Adverse drug reaction (ADR).** Is a response to a drug which is noxious and unintended, and may occur during treatment.

**Co-morbidity.** Is an illness which was diagnosed together with MBR-TB, but different from ADR.

**Mortality (death).** A patient is considered dead and counted as an event if and only if it was documented in the patient's discharge summary sheet and confirmed by the signature of the authorized physician.

## Results

### Characteristics of the study population

Of 213 records screened for eligibility, 13 records were excluded and 200 MDR-TB patients' records were included in the analysis (Fig 1).

### Socio-demographic characteristics

The majority, 108(54%) of the patients were males. The mean ± standard deviation (SD) age of the study participants was 32.9±9.5years. The largest proportions, 78 (39%) of the participants were Muslims. Most, 111 (55.5%) of them were from rural areas and 99 (49.5%) of the partici-pants were married. Seventy-four (37%) patients had attained secondary level of education. About 62 (31%) of the study participants were merchants. Non-smokers and non-alcoholics comprised 190 (95%) and 172 (86%), respectively. The baseline smoking status, alcohol consumption and body mass index were associated with patients' status (p<0.05) (Table 1).

### Clinical characteristics and drug-related variables

Pulmonary tuberculosis (187/200) was commonly diagnosed among the study participants. The majority, 156 (78%) of the MDR-TB cases were previously treated/relapse. Thirteen

**Table 1. Socio-demographic characteristics of MDR-TB patients treated at MDR-TB centers of South and Southwest of Ethiopia, April 14 to May 14, 2019.**

| Variables | Category | Frequency | Patient status (n = 200) | | p-value |
|---|---|---|---|---|---|
| | | | Non-survivors (n = 22) | Survivors (n = 178) | |
| Sex | Male | 108(54%) | 13(59%) | 95(53%) | 0.612 |
| | Female | 92(46%) | 9(41%) | 83(47%) | |
| Residence | Urban | 89(44.5) | 10(45.5%) | 79(44%) | 0.924 |
| | Rural | 111(55.5%) | 12(54.5%) | 99(56%) | |
| Smoking status | Yes | 10(5%) | 4(18%) | 6(3.4%) | 0.003* |
| | No | 190(95%) | 18(82%) | 172(96.6%) | |
| Alcoholic status | Yes | 28(14%) | 9(41%) | 19(10.7%) | p<0.001* |
| | No | 172(86%) | 13(59%) | 159(89.3%) | |
| Marital status | Single | 88(44%) | 9(41%) | 79(44%) | 0.587 |
| | Married | 99(49.5%) | 12(54%) | 88(49%) | |
| | Divorce | 10(5%) | 1(2%) | 9(5%) | |
| | Widowed | 3(1.5%) | 1(2%) | 2(2%) | |
| Age | <25 | 50(25%) | 4(18%) | 46(25.5%) | 0.179 |
| | 26–45 | 134(67%) | 15(68%) | 124(70%) | |
| | >45 | 16(8%) | 3(14%) | 8(4.5%) | |
| BMI | <18.5 | 55(27.5%) | 10(45.5%) | 45(25%) | 0.046* |
| | >18.5 | 145(72.5%) | 12(54.5%) | 133(75%) | |

*Statistically significant difference at *p<0.05*.

**Table 2. Baseline clinical and drug-related characteristics of MDR-TB patients treated at MDR-TB treatment centers of South and Southwest Ethiopia, April 14 to May 14, 2019.**

| Variables | Category | Frequency | Patient status (n = 200) | | p-value |
|---|---|---|---|---|---|
| | | | Non-survivors (n = 22) | Survivors (n = 178) | |
| Site of disease | Pulmonary | 187(93.5%) | 19(86%) | 168(94%) | 0.150 |
| | Extra-pulmonary | 13(6.5%) | 3(14%) | 10(6%) | |
| Treatment group | New | 22(11%) | 3(14%) | 19(10.7%) | 0.377 |
| | Previously treated | 156(78%) | 19(86%) | 137(77%) | |
| | After loss to follow-up | 9(4.5%) | 0(0%) | 9(5%) | |
| | After treatment failure | 13(6.5%) | 0(0%) | 13(7.3%) | |
| Comorbidity | Yes | 56(28%) | 20(90.9%) | 36(20%) | p<0.001* |
| | No | 144(72%) | 2(9.1%) | 142(80%) | |
| HIV sero-status | Sero-positive | 44(22%) | 14(63.6% | 30(16.9%) | p<0.001* |
| | Sero-negative | 156(78%) | 8(36.4%) | 148(83.1%) | |
| Diabetes | Yes | 9 (4.5%) | 4(18.2%) | 5(2.8%) | 0.001* |
| | No | 191(95.5%) | 18(81.8%) | 173(97.2%) | |
| AKI | Yes | 7(3.5%) | 5(22.7%) | 2(1.1%) | p<0.001* |
| | No | 193(96.5%) | 17(77.3%) | 176(98.9%) | |
| Adherence status | Good | 173(86.5%) | 12(54.5%) | 161(90.5%) | p<0.001* |
| | Fair | 20(10%) | 6(27%) | 14(8%) | |
| | Poor | 7(3.5%) | 4(18.5%) | 3(1.5%) | |
| Taking Vit B6 | Yes | 196(98%) | 20(90.9%) | 176(99%) | 0.012* |
| | No | 4(2%) | 2(9.1%) | 2(1%) | |
| Adverse drug reaction | Yes | 57(28.5%) | 17(77.3%) | 40(22.5%) | p<0.001* |
| | No | 143(71.5%) | 5(22.7%) | 138(77.5%) | |
| Ethambutol (E) | Resistance | 18(9%) | 4(18.5%) | 14(8%) | 0.111 |
| | Susceptible | 182(91%) | 18(81.5%) | 164(92%) | |
| Streptomycin (S) | Resistance | 11(5.5%) | 4(18.5%) | 7(4%) | 0.006* |
| | Susceptible | 189(94.5%) | 18(81.5%) | 171(96%) | |
| Levofloxacin (Lfx) | Resistance | 4(2%) | 1(4.5%) | 3(2%) | 0.366 |
| | Susceptible | 196(98%) | 21(95.5%) | 175(98%) | |

AKI: Acute kidney injury, HIV: Human immune virus, *Statistically significant at p-value <0.05.

(6.5%) patients had treatment after failure, 22 (11%) were new MDR-TB cases and 9 (4.5%) were after loss to follow-up. Fifty-six (26.5%) patients had comorbidity. All patients were tested for HIV infection and 44 (22%) patients were found to be HIV positive. Diabetes mellitus 9 (4.5%) and acute kidney injury 7 (3.5%) were among the common comorbidities.

On the drug sensitivity test, samples of 126 (63%) patients were resistant to Isoniazid (INH) and 100% of the patients were resistant to rifampicin (RIF). Furthermore, 18 (9%) patients were resistant to Ethambutol. Whereas, 11 (4.5%) and 4 (2%) patients were resistant to Streptomycin and Levofloxacin, respectively (Table 2).

The median (IQR) hemoglobin and thyroid-stimulating hormone level of the study participants were 14 (13–15) g/dl and 5 (4–6) μ/ml, respectively. Similarly, the median (IQR) Serum Creatinine and alanine aminotransferase level were 0.87 (0.57–0.98) mg/dl and 34 (27–41) IU/L, respectively. As, described in Table 2, the presence of comorbidity, experiencing adverse drug reactions, adherence status and resistance to Streptomycin were associated with the patient's status (p<0.05).

## Patient survival and its predictors

In this study, the overall analysis time at risk and under observation was 8,185days. During the study period, 22 (11%) deaths were reported. The incidence density of death among the study population was 11.99; 95% CI [7.89–18.21] per 100,000person-years. The first death was recorded at 180-days after the treatment initiation and the over all median (interquartile range (IQR)) survival time to death was 375 (249–457) days. The median (IQR) survival times corresponding to the MDR-TB treatment regemens namely; (Z,E,Cm,Lfx,Eto,Cs), (Z,Cm,Lfx,Eto, Cs) and (Z,Cm,Lfx,Pto, Cs) were 306 (215.5–418), 367 (275–398) and 414 (365–457) days respectively. There was no statistically significant difference in survival (log-rank p = 0.54) among MDR-TB treatment regimens (Fig 2).

Furthermore, the cox proportional hazard regression model was fitted to identify predictors of patients' survival (mortality). Accordingly, low hemoglobin level, ADR, comorbidity, smoking, alcoholic status, adherence status and exposure to Vitamin B6 were significantly associated with patients' survival (p<0.25). On multivariate cox-regression, co-morbidity, alcoholic status, and poor adherence status were independently associated with survival. Hence, patients with comorbidity had 23 times higher hazards of death (AHR = 23.68, 95% CI [4.85–115.46]]). Similarly, patients who consume alcohol had 4.5 times higher hazards of death (AHR = 4.53, 95% CI [1.21–16.97]. Poor adherence was also responsible for increased risk of mortality by more than 12 times (AHR = 12.27, 95% CI [2.83–53.21] (Table 3).

## Discussion

This study summarized the patients' survival and its predictors among MDR-TB patients who were treated at MDR-TB treatment centers of south and southwest Ethiopia.

**Table 3. Crude and adjusted Cox-proportional hazard regression model for predictors of moratlity (survival) among MDR-TB patients treated at MDR-TB treatment centers of South and Southwest Ethiopia, April 14 to May 14, 2019.**

| Variables | | CHR [95%CI] | P-value | AHR [95%CI] | p-value |
|---|---|---|---|---|---|
| Smoker | Yes | 5.22[1.76–15.53] | 0.003 | 1.26[0.21–7.91] | 0.80 |
| | No | 1.00 | | 1.00 | |
| Comorbidity | Yes | 38.34[8.89–165.27] | p<0.001 | 23.68[4.85–115.46] | p<0.001 |
| | No | 1.00 | | 1.00 | |
| Alcoholic users | Yes | 5.03[2.15–11.80] | p<0.001 | 4.53[1.21–16.97] | 0.03 |
| | No | 1.00 | | 1.00 | |
| Adherence | Good | 1.00 | | 1.00 | |
| | Fair | 6.29[2.34–17.01] | p<0.001 | 1.40[0.40–4.94] | 0.60 |
| | Poor | 14.24[4.51–44.91] | p<0.001 | 12.27[2.83–53.21] | 0.001 |
| Receiving Vitamin B6 | Yes | 0.09[0.02–0.41] | 0.002 | 0.40[0.06–2.83] | 0.36 |
| | No | 1.00 | | 1.00 | |
| ADR | Yes | 9.45[3.48–25.67] | p<0.001 | 2.55[0.79–8.20] | 0.12 |
| | No | 1.00 | | 1.00 | |
| Ethambutol (E) | Resistance | 2.42[0.82–7.17] | 0.11 | 1.56[0.41–5.88] | 0.51 |
| | Susceptible | 1.00 | | 1.00 | |
| BMI | <18.5 | 2.19[0.95–5.08] | 0.07 | 0.52[0.16–1.63] | 0.26 |
| | >18.5 | 1.00 | | 1.00 | |
| HGB | <12.5gm/dl | 2.23[0.75–6.58] | 0.15 | 1.76[0.39–7.91] | 0.46 |
| | >12.5gm/dl | 1.00 | | 1.00 | |

Good adherence (≥95%), Fair adherence (85–94%), Poor adherence (>85%), HGB: Haemoglobin, ADR: Adverse drug reaction.

The study found 22 (11%) of death. The incidence density of mortality was 11.99, 95% CI [7.89–18.21] per 100,000person-years. The overall median (IQR) survival time to death was 375 (249–457) days. There was no statistically significance in terms of survival among MDR-TB regimens (p = 0.54). The Cox regression analysis revealed that the presence of co-morbidity, alcohol consumption and poor adherence were independent predictors of survival (death).

The overall incidence density of mortality in the current study was comparable with a study conducted in Lithuania that reported 11 per 100,000 person-per years [16]. But lower than the study by Girum et al [17] in which the reported incidence density of death was 7per 100per-son-years. The differences might be due to the inclusion of a small number of patients (154 versus 200) and a shorter follow-up period of the former study. The reported death rate (11%) in our study was also lower than the study conducted in India 17%, South Africa 20%, and eastern Ethiopia 18.3% [11, 18, 19], but higher than the finding from Peru, 5% [8].

In this study, the overall median (IQR) survival time to death was 375 (249–457) days. The finding was lower than the study from central Ethiopia which reported the median survival time of 480days [11]. But much lower median (IQR) survival time to death was reported by *Kamban et al* [10], i.e. 78(33.3–154.5) days. The finding regarding the median (IQR) survival time associated with each MDR-TB regimens lacks statistical significane, but the regimen, (Z, Cm,Lfx,Pto, Cs) was associated with improved median (IQR) survival time,.i.e. 414 (365–457) days.

In the current study, patients with comorbidity had 24 times higher hazard of death (AHR = 23.68, 95% CI [4.85–115.46]. HIV/AIDS, diabetes mellitus and acute kidney injury were the common comorbidities. The presence of such comorbidities, mainly HIV/AIDS [6, 9] and diabetes mellitus were strongly related to immune-suppression. Tuberculosis facilitates HIV replication and viral diversification rates through proinflammatory cytokine production. Proinflammatory cytokines in turn increase HIV viral replication and diversity, hence facilitating immune-suppression [20]. A study by *Chung-Degado et al* indicted patients with comorbidity had 5.4 times higher hazard of death [8]. Our finding was also concurrent with the previous studies conducted in South Africa [5], Tanzania [6], and Ethiopia [7].

This study identified alcohol consumption as another predictor of death. Patients who drank alcohol had 4.7 times higher hazards of death (AHR = 4.53, 95% CI [1.21–16.97]). Alcohol can amplify adverse drug reaction including hepatic toxicity. Moreover, Alcohol consumption detracts general health and may impair immune responses against *M. tuberculosis* [21]. Concurrent findings were also reported by *Duraisamy et al*, in which persons who consumed alcohol during treatment had 4.3 times higher hazard of death (AHR = 4.3, 95% CI [1.1–17.6] [22]. Studies also indicated that an estimated 10% of tuberculosis (TB) deaths were attributable to alcohol use globally [23]. In this study, poor adherence increased the risk of death by 12.3 times (AHR = 12.27, 95% CI [2.83–53.21]. *Habteyes et.al* found treatment interruption was significantly associated with unsuccessful treatment outcomes (ARR = 1.9; 95% CI [1.4–2.6]) [24].

This study was not without limitations. Firstly, The retrospective nature of the data source limited us from tracking the major causes of death. Secondly, the method of patients' adherence assessment was also subjective as it is based on patient self report. Third, most of the patients have no data on sputum smear microscopy results. Lastly, but not least, missing patients' income status, wider confidence intervals and inability to screen out the exact causes of death were some of the major hiccups of this study.

## Conclusion

In conclusion, this study found a high rate of mortality among patients receiving MDR-TB treatment. Alcohol use, poor adherence, and the presence of co-morbidity were independent predictors of death. This study provided insight into how to provide optimal care of MDR-TB patients with comorbidities, poorly adhered to therapy and habit of alcohol use. However, given all the limitations mentioned above, we urge the readers to interpret the findings of this study cautiously.

## Supporting information

**S1 File.**
(RAR)

## Acknowledgments

The authors thank the data collectors and all staff members of the study settings for their valuable contribution. We also would like to thank Jimma University for providing this opportunity to conduct this research.

## Author Contributions

**Conceptualization:** Asnake Balche Bade, Teshale Ayele Mega.

**Data curation:** Asnake Balche Bade, Teshale Ayele Mega.

**Formal analysis:** Asnake Balche Bade, Teshale Ayele Mega.

**Investigation:** Teshale Ayele Mega.

**Resources:** Teshale Ayele Mega.

**Software:** Asnake Balche Bade, Teshale Ayele Mega.

**Supervision:** Teshale Ayele Mega.

**Writing – original draft:** Teshale Ayele Mega.

**Writing – review & editing:** Asnake Balche Bade.

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
