## [Decision Letter · Decision Letter 0]

15 Jun 2020

PONE-D-20-08138

Mortality and its predictors among patients receiving Multi-drug resistance tuberculosis treatment in Ethiopia: Multicenter observational study

PLOS ONE

Dear Dr. Ayele,

Thank you for submitting your manuscript to PLOS ONE. After careful consideration, we feel that it has merit but does not fully meet PLOS ONE’s publication criteria as it currently stands. Therefore, we invite you to submit a revised version of the manuscript that addresses the points raised during the review process.

We look forward to receiving your revised manuscript.

Kind regards,

Ram Chandra Bajpai, Ph.D.

Academic Editor

PLOS ONE

Journal Requirements:

Reviewers' comments:

Reviewer's Responses to Questions

**Comments to the Author**

1. Is the manuscript technically sound, and do the data support the conclusions?

Reviewer #1: No

Reviewer #2: Yes

2. Has the statistical analysis been performed appropriately and rigorously? 

Reviewer #1: No

Reviewer #2: Yes

3. Have the authors made all data underlying the findings in their manuscript fully available?

Reviewer #1: Yes

Reviewer #2: Yes

4. Is the manuscript presented in an intelligible fashion and written in standard English?

Reviewer #1: No

Reviewer #2: Yes

5. Review Comments to the Author

Reviewer #1: This study lacks clarity what they intended to measure in the methodology section; they should have defined the study outcomes/outcomes clearly. They didn’t indicated what is mortality? mortality due to…? Including the time when the deaths occurred and how they were confirmed the mortality is due to MDR TB or due to something else to avoid computing risks for the deaths. In addition, as the study is survival, it seems that study somewhat misplaced from its intended objectives as the aim of the study should have been to estimate the time to death (time to event).

One of the futures of institutional based datasets including hospitals is known with missing. What was the level of data missing in your case? How you managed the data missing? Have you checked whether the missing cases were not different from the included cases? The manuscript doesn’t provide any information on how you handled data missing.

Data coming from different zones or regions or even hospital can have a peculiar nature, patients from the same zone or hospital tends to share the same nature and patients or data from different zones or hospital clearly tends to show different nature than within the zone or hospital. How did you manage this nature of homogeneity within and heterogeneity between hospital properties?

The authors didn’t indicate why they failed to fit frailty model rather they conducted just cox proportional hazard model which assumes there is no variability between groups, in this case may be hospitals. However, you conducted a multicenter retrospective study (Butajira, Arbaminch and Shenengibe General Hospitals).

The authors didn’t operationalize some important terminologies example Comorbidity, Adherence status (what is good adherence, fair, poor adherence ???) based on what ???, what by mean adverse drug reaction mean ?? alcohol use??

Page 17, table 2: the authors mentioned that “…. MDR-TB patients at selected

MDR-TB centers of Ethiopia…”, why selected centers in Ethiopia? These mentioned centers are the very small pieces of hospitals in South nations and nationalities live alone Ethiopia. So please properly edit

In this study the authors mentioned in the abstract and last paragraph sections of introduction section that they aimed to study mortality and its predictors among patients receiving multi-drug resistance tuberculosis …., however, when you see the method section (data analysis design) doesn’t seem to support these objectives.

It seems that you need to fit logistic regression and need to find predictors of morality, but they conducted survival model. So, the objective they set, and data analysis design and the findings of the study are not support each other. Need to make serious corrections.

Is the assumptions of cox-proportional hazard fulfilled? didn’t see in the result section and even there is nothing in the method section which assumptions to be used? The confidence interval for the cox-proportional hazard (Comorbidity, Alcoholic users, adherence) are very wide, see table 3 results, what could be the main reasons? Do you think the results are valid?

The way this manuscript written was very floppy, the introduction part didn’t show clearly what gap this study intended to fill, and the objectives are poorly or inappropriately set. The primary aim of survival analysis is to identify the mean or median survival time for the death, this should be placed in the objective part and should get emphasis throughout the study.

The method section has serious limitations, even didn’t clearly indicate how the outcome of the study (death) would be assessed. Didn’t show clearly whether death is due to MDR-TB or not. Surprisingly, in the discussion section, the authors indicated that the limitation of the study is difficulty to determine exact causes of death”! as a study, the investigators expected clearly identify the cause of deaths, and should include only confirmed deaths due to MDR TB. Should have shown the study participants with the outcome(death) selected and included in the analysis in order to find the estimate of the outcome (incidence of death due to MDR TB).

The result and the discussion sections were also poorly written, and not properly discussed.

Reviewer #2: In this manuscript authors develop the assess mortality and its predictors among patients receiving multi-drug

resistance tuberculosis treatment at selected MDR TB treatment centers of southern and

southwestern Ethiopia. The manuscript written the nicely. Minor modifications are required and those are highlighted in this attached file.

6. PLOS authors have the option to publish the peer review history of their article (what does this mean?). If published, this will include your full peer review and any attached files.

Reviewer #1: No

Reviewer #2: No

---

## [Author Response · Author response to Decision Letter 0]

17 Sep 2020

Rebuttal letter

Dear editor below is the response given to the concerns raised by the reviewers. We fully acknowledge reviewers effort and considered the comments as much as possible. 

Reviewer #1: This study lacks clarity what they intended to measure in the methodology section; they should have defined the study outcomes/outcomes clearly. They didn’t indicated what is mortality? Mortality due to…? Including the time when the deaths occurred and how they were confirmed the mortality is due to MDR TB or due to something else to avoid computing risks for the deaths. In addition, as the study is survival, it seems that study somewhat misplaced from its intended objectives as the aim of the study should have been to estimate the time to death (time to event).

Response: We defined mortality, indicated its possible cause and how it was confirmed. Moreover, we modified figure 2 and include additional information about survival times (see “The outcome definition and measurement” under methods section and figure 2). 

One of the futures of institutional based datasets including hospitals is known with missing. What was the level of data missing in your case? How you managed the data missing? Have you checked whether the missing cases were not different from the included cases? The manuscript doesn’t provide any information on how you handled data missing.

Response: Luckily, the MDR-TB centres in Ethiopia are fully automated, guided centrally, and strictly supervised by the federal ministry of health (FMoH). Each patient data is computerized and automatically detected centrally by help of Ethiopian telecommunication when an MDR-TB is detected. The settings are also well equipped for this purpose. So, it is less likely to have missing data as that of other services, but still there were some gaps regarding data handling and as we have reported in fig 1 we excluded 3charts with missed follow-up (outcome) data. Surely, these charts were not different from the included cases.

Data coming from different zones or regions or even hospital can have a peculiar nature, patients from the same zone or hospital tends to share the same nature and patients or data from different zones or hospital clearly tends to show different nature than within the zone or hospital. How did you manage this nature of homogeneity within and heterogeneity between hospital properties?

Response: we consider the data as homogenous because the study settings are at similar level of care (general hospitals). More importantly, the facilities are centrally guided and controlled by a single organization, despite the difference in location. The health care professional receive similar training, while introducing in to the services, guidelines, formats and updates will be provided for the settings uniformly. 

The authors didn’t indicate why they failed to fit frailty model rather they conducted just cox proportional hazard model which assumes there is no variability between groups, in this case may be hospitals. However, you conducted a multicenter retrospective study (Butajira, Arbaminch and Shenengibe General Hospitals).

Reponses: We didn’t fit frailty model because of the reason mentioned above. 

The authors didn’t operationalize some important terminologies example Comorbidity, Adherence status (what is good adherence, fair, poor adherence ???) based on what ???, what by mean adverse drug reaction mean ?? alcohol use??

Response: Operationalized!

Page 17, table 2: the authors mentioned that “…. MDR-TB patients at selected

MDR-TB centers of Ethiopia…”, why selected centers in Ethiopia? These mentioned centers are the very small pieces of hospitals in South nations and nationalities live alone Ethiopia. So please properly edit

Response: the term “selected” is used because we have other hospitals left uncovered due to logistic issue. Anyhow we have expressed it differently. 

In this study the authors mentioned in the abstract and last paragraph sections of introduction section that they aimed to study mortality and its predictors among patients receiving multi-drug resistance tuberculosis …., however, when you see the method section (data analysis design) doesn’t seem to support these objectives. It seems that you need to fit logistic regression and need to find predictors of morality, but they conducted survival model. So, the objective they set, and data analysis design and the findings of the study are not support each other. Need to make serious corrections.

Response: correction is made.

Is the assumptions of cox-proportional hazard fulfilled? didn’t see in the result section and even there is nothing in the method section which assumptions to be used? 

Response: mentioned under the data analysis section

The confidence interval for the cox-proportional hazard (Comorbidity, Alcoholic users, adherence) are very wide, see table 3 results, what could be the main reasons? Do you think the results are valid?

Response: the confidence intervals (CI) were wide because of the smaller number of events included in the analysis. Regarding the validly of the finding, we mentioned under our limitation to give insight for the readers for cautious interpretation of the finding. The wider the CI doesn’t mean that the findings were invalid indeed, but it hints you that the events or sample size was smaller than the required. As you can see, the number of active patients in the included settings was also small, the reason why we included all patients. 

The way this manuscript written was very floppy, the introduction part didn’t show clearly what gap this study intended to fill, and the objectives are poorly or inappropriately set. The primary aim of survival analysis is to identify the mean or median survival time for the death, this should be placed in the objective part and should get emphasis throughout the study.

Response: We have fully revised the manuscript including introduction of new terminologies

The method section has serious limitations, even didn’t clearly indicate how the outcome of the study (death) would be assessed. Didn’t show clearly whether death is due to MDR-TB or not. Surprisingly, in the discussion section, the authors indicated that the limitation of the study is difficulty to determine exact causes of death”! as a study, the investigators expected clearly identify the cause of deaths, and should include only confirmed deaths due to MDR TB. Should have shown the study participants with the outcome (death) selected and included in the analysis in order to find the estimate of the outcome (incidence of death due to MDR TB).

Response: Dear reviewer, we would like to remind you that, as a secondary data user, we cannot go far beyond what has been described by the caring physicians. This is actually the limitation of secondary data which we have mentioned under the limitation section. It is quite difficult to conclude all deaths could be perfectly due TB. For example, if a patient dies of acute renal injury due to drug adverse effects this is not due to TB. But as long as he/she was an MDR-TB patient the physician could record him/her as died of TB and we can’t disregard this patient from the analysis. So, that why we mentioned this as one of our limitations. 

The result and the discussion sections were also poorly written, and not properly discussed.

Response: We have revised the whole manuscript

Reviewer #2: In this manuscript authors develop the assess mortality and its predictors among patients receiving multi-drug

resistance tuberculosis treatment at selected MDR TB treatment centers of southern and

southwestern Ethiopia. The manuscript written the nicely. Minor modifications are required and those are highlighted in this attached file.

Response: Thank you indeed. All your feedbacks were considered

---

## [Decision Letter · Decision Letter 1]

20 Oct 2020

Survival status and its predictors among Multi-drug resistance tuberculosis treated patients in Ethiopia: Multicenter observational study

PONE-D-20-08138R1

Dear Dr. Mega,

We’re pleased to inform you that your manuscript has been judged scientifically suitable for publication and will be formally accepted for publication once it meets all outstanding technical requirements.

Kind regards,

Ram Chandra Bajpai, Ph.D.

Academic Editor

PLOS ONE

Additional Editor Comments (optional):

Reviewers' comments:

Reviewer's Responses to Questions

**Comments to the Author**

1. If the authors have adequately addressed your comments raised in a previous round of review and you feel that this manuscript is now acceptable for publication, you may indicate that here to bypass the “Comments to the Author” section, enter your conflict of interest statement in the “Confidential to Editor” section, and submit your "Accept" recommendation.

Reviewer #1: All comments have been addressed

Reviewer #2: (No Response)

2. Is the manuscript technically sound, and do the data support the conclusions?

Reviewer #1: Yes

Reviewer #2: Yes

3. Has the statistical analysis been performed appropriately and rigorously? 

Reviewer #1: Yes

Reviewer #2: Yes

4. Have the authors made all data underlying the findings in their manuscript fully available?

Reviewer #1: Yes

Reviewer #2: Yes

5. Is the manuscript presented in an intelligible fashion and written in standard English?

Reviewer #1: Yes

Reviewer #2: Yes

6. Review Comments to the Author

Reviewer #1: (No Response)

Reviewer #2: Authors have incorporated all the comments in revised version. I accept this manuscript and recommend for publication

7. PLOS authors have the option to publish the peer review history of their article (what does this mean?). If published, this will include your full peer review and any attached files.

Reviewer #1: No

Reviewer #2: No

---

## [Editor Report · Acceptance letter]

28 Oct 2020

PONE-D-20-08138R1 

Survival status and its predictors among Multi-drug resistance tuberculosis treated patients in Ethiopia: Multicenter observational study 

Dear Dr. Mega:

I'm pleased to inform you that your manuscript has been deemed suitable for publication in PLOS ONE. Congratulations! Your manuscript is now with our production department. 

Kind regards, 

on behalf of

Dr. Ram Chandra Bajpai 

Academic Editor

PLOS ONE